# Diagnostic Performance of Magnetic Resonance Imaging for Parathyroid Localization of Primary Hyperparathyroidism: A Systematic Review

**DOI:** 10.3390/diagnostics14010025

**Published:** 2023-12-22

**Authors:** Max H. M. C. Scheepers, Zaid Al-Difaie, Lloyd Brandts, Andrea Peeters, Bjorn Winkens, Mahdi Al-Taher, Sanne M. E. Engelen, Tim Lubbers, Bas Havekes, Nicole D. Bouvy, Alida A. Postma

**Affiliations:** 1GROW School for Oncology and Developmental Biology, Maastricht University, 6229 ER Maastricht, The Netherlands; 2Department of Clinical Epidemiology and Medical Technology Assessment, Maastricht University Medical Centre, 6202 AZ Maastricht, The Netherlands; 3Department of Methodology and Statistics, CAPHRI, Maastricht University Medical Centre, 6229 HX Maastricht, The Netherlands; bjorn.winkens@maastrichtuniversity.nl; 4Department of Surgery, Maastricht University Medical Center, 6202 AZ Maastricht, The Netherlandssanne.engelen@mumc.nl (S.M.E.E.); 5Department of Internal Medicine, Division of Endocrinology and Metabolic Disease, Maastricht University Medical Center, 6202 AZ Maastricht, The Netherlands; 6NUTRIM School of Nutrition and Translational Research in Metabolism, Maastricht University, 6229 ER Maastricht, The Netherlands; 7Department of Radiology and Nuclear Medicine, School for Mental Health and Sciences (MHENS), Maastricht University Medical Center, 6202 AZ Maastricht, The Netherlands; l.jacobi@mumc.nl

**Keywords:** magnetic resonance imaging, primary hyperparathyroidism, diagnostics, preoperative

## Abstract

Accurate preoperative localization is crucial for successful minimally invasive parathyroidectomy in primary hyperparathyroidism (PHPT). Preoperative localization can be challenging in patients with recurrent and/or multigland disease (MGD). This has led clinicians to investigate multiple imaging techniques, most of which are associated with radiation exposure. Magnetic resonance imaging (MRI) offers ionizing radiation-free and accurate imaging, making it an attractive alternative imaging modality. The objective of this systematic review is to provide an overview of the diagnostic performance of MRI in the localization of PHPT. PubMed and Embase libraries were searched from 1 January 2000 to 31 March 2023. Studies were included that investigated MRI techniques for the localization of PHPT. The exclusion criteria were (1) secondary/tertiary hyperparathyroidism, (2) studies that provided no diagnostic performance values, (3) studies published before 2000, and (4) studies using 0.5 Tesla MRI scanners. Twenty-four articles were included in the systematic review, with a total of 1127 patients with PHPT. In 14 studies investigating conventional MRI for PHPT localization, sensitivities varied between 39.1% and 94.3%. When employing more advanced MRI protocols like 4D MRI for PHPT localization in 11 studies, sensitivities ranged from 55.6% to 100%. The combination of MR imaging with functional techniques such as 18F-FCH-PET/MRI yielded the highest diagnostic accuracy, with sensitivities ranging from 84.2% to 100% in five studies. Despite the limitations of the available evidence, the results of this review indicate that the combination of MR imaging with functional imaging techniques such as 18F-FCH-PET/MRI yielded the highest diagnostic accuracy. Further research on emerging MR imaging modalities, such as 4D MRI and PET/MRI, is warranted, as MRI exposes patients to minimal or no ionizing radiation compared to other imaging modalities.

## 1. Introduction

The definitive treatment of primary hyperparathyroidism (PHPT) requires surgical resection of hyperfunctioning parathyroid glands [1,2]. Accurate preoperative localization of parathyroid lesions is crucial to achieving a successful minimally invasive parathyroidectomy [3,4]. Traditionally, preoperative localization has been performed with ultrasonography (US) and/or sestamibi scintigraphy (MIBI), which, in most PHPT cases, has a high diagnostic accuracy. However, in patients with multigland disease (MGD), e.g., multiple adenomas or four-gland hyperplasia, and recurrent disease, these imaging modalities can be inaccurate [5,6,7]. The incidence of MGD is reported to be up to 15–20% of PHPT cases [8,9]. Negative or inconclusive pre-operative imaging may necessitate bilateral neck exploration to find the parathyroid lesion, which increases the risk of perioperative complications such as transient/permanent hypoparathyroidism and recurrent laryngeal nerve injury [10,11,12].

This has led clinicians to investigate and utilize other emerging imaging techniques. CT has been studied extensively for the localization of PHPT [4,13,14]. One option, 4D-CT, is an innovative CT technique that uses multiple phases of contrast-enhanced scanning to visualize the unique perfusion characteristics of parathyroid lesions [15]. Several studies have found a superior diagnostic accuracy of 4D-CT compared to US and MIBI [15,16,17,18,19]. However, a drawback of 4D-CT is the relatively high radiation exposure that results from the multiple scanning phases [20]. This has raised concerns regarding a possible increased risk of thyroid cancer, particularly in younger patients [20]. Alternatively, radiolabeled choline positron emission tomography CT is emerging as a highly promising imaging modality in the localization of PHPT [21,22,23,24]. Radiolabeled PET imaging has mainly been investigated in combination with CT, and most studies revealed a superior diagnostic accuracy of ^18^F-Flourocholine-PET/CT (^18^F-FCH-PET/CT) compared to US and MIBI [21,22,23,24]. An added advantage of choline PET-CT is that it exposes patients to similar or less ionizing radiation than MIBI [25].

Magnetic resonance imaging (MRI) offers ionizing radiation-free and accurate imaging, making it an attractive alternative to CT and MIBI. Due to recent advancements, including the use of 3T magnets and dynamic contrast-enhanced MRI (4D MRI) protocols, diagnostic accuracies have improved and acquisition times have been reduced [26,27]. Using 4D MRI, radiologists are now able to utilize multiple post-contrast phases to assess the unique contrast perfusion characteristics of parathyroid lesions, similar to 4D-CT [27]. Unlike 4D-CT, 4D MRI eliminates the need for ionizing radiation exposure, enabling additional post-contrast phase scans and improving perfusion analysis. Moreover, the integration of PET techniques with MRI, known as PET/MRI, could offer additional advantages compared to choline PET-CT [24,28]. This additional advantage is attributed to a further reduction in radiation exposure and optimal soft-tissue contrast in the neck [29,30,31].

To the best of our knowledge, there is currently no systematic review on the diagnostic performance of MRI in the localization of PHPT. The aim of this systematic review is to provide an overview of the diagnostic accuracy of several MR-imaging techniques for the localization of PHPT.

## 2. Materials and Methods

### 2.1. Search Strategy

On 31 March 2023, an electronic search was performed in the PubMed and Embase databases. Keywords, including hyperparathyroidism, parathyroid glands, adenoma, magnetic resonance imaging, sensitivity, and specificity, were used as search terms, combined using AND/OR operators. The full search strategy can be found in Appendix A. This study followed the reporting guidelines of the Preferred Reporting Items for Systematic Reviews and Meta-analyses (PRISMA) [32]. The protocol was not registered, as it is not obligatory, according to the PRISMA statement.

A total of 473 articles were found. After removal of duplicates, 372 articles were screened for eligibility by reading the title and abstract. A total of 153 articles were screened independently for eligibility by 2 reviewers (M.H.M.C.S. and Z.A) who read the full text. Any discrepancies were resolved in a consensus meeting. Diagnostic studies were included that investigated any MRI technique for PHPT localization. The exclusion criteria were (1) secondary/tertiary hyperparathyroidism, (2) studies that did not provide diagnostic performance values, (3) studies published before 2001, and (4) studies using 0.5 Tesla (0.5T) magnet MRI scanners (1.5 T and 3 T have become the accepted standard of MRI scanners worldwide [33,34]). 

Two independent reviewers (M.H.M.C.S. and Z.A.) collected the following information and compiled it into tables based on the included articles: initial surgery for PHPT or surgery for recurrent/persistent PHPT, number of patients with single gland disease (SGD) and multigland disease (MGD), type of (PET/) MRI scanner, reference test, sensitivity, specificity, negative predictive value (NPV), positive predictive value (PPV), and accuracy. Furthermore, the definition used for the diagnostic values calculated in the included studies was gathered. Lesion-based sensitivity refers to the proportion of actual parathyroid adenomas that are correctly identified as positive by the diagnostic test. True positive (for lesion-based) is defined as correct localization on MRI, where the resected specimen was confirmed to be a parathyroid adenoma by pathology reports. Therefore, lesion-based sensitivity was based on the pathology reports. Patient-based sensitivity refers to the proportion of patients with true positive results out of all the patients who actually have parathyroid adenomas. True positive (for patient-based) is defined as correct localization on MRI, which led to curation after resection of the localized adenomas. Therefore, patient-based sensitivity was based on the MRI-guided surgery results.

In this systematic review, a study was deemed a head-to-head comparative study if it investigated two distinct imaging modalities for localizing primary hyperparathyroidism (PHPT) in the same group of patients. This implies that each patient underwent both imaging modalities, and the results were subsequently validated by the same pathology report.

### 2.2. Risk of Bias Assessment

Risk of bias was assessed by the Quality Assessment of Diagnostic Studies 2 tool (QUADAS-2) [35]. Assessment was performed independently by 2 reviewers (M.H.M.C.S. and Z.A). Any discrepancies were resolved in a consensus meeting. The QUADAS-2 tool can be found in Appendix A.

We opted not to conduct a meta-analysis in this study due to the amount of heterogeneity in the imaging protocols employed across the included studies. The variations in imaging techniques and protocols among the selected studies precluded the possibility of pooling the data for a meaningful quantitative analysis.

## 3. Results

The literature search identified 24 publications [26,36,37,38,39,40,41,42,43,44,45,46,47,48,49,50,51,52,53,54,55,56,57,58] that met the inclusion criteria for the systematic review. Figure 1 shows the flow diagram of the identification, screening, eligibility, and selection process. The included studies investigated 4D MRI (*n* = 7) [26,36,41,44,49,50,52], conventional MRI (*n*  = 9) [37,38,42,43,51,54,55,56,58], ^18^F-Flourocholine-PET/MRI (^18^F-FCH-PET/MRI) (*n* = 2) [46,57], both 4D- and conventional MRI (*n* = 3) [40,48,53], ^18^F-FCH-PET/MRI and conventional MRI (*n* = 2) [39,45], and ^18^F-FCH-PET/MRI and 4D MRI (*n* = 1) [47], with a total of 1127 patients with PHPT across all studies. A total of 11 studies were retrospective studies [36,39,40,43,45,48,49,53,54,55,56], and 13 were prospective studies [26,37,38,41,42,44,46,47,50,51,52,57,58]. Study characteristics are summarized in Table 1.

Twenty studies included only patients who underwent initial surgery for PHPT [26,36,37,38,39,41,42,44,45,46,47,48,49,50,51,52,53,54,55,58], two studies included only recurrent PHPT patients with prior (para-) thyroid surgery [40,43], and two studies included both [48,57]. All studies used histopathological analysis for diagnostic confirmation of parathyroid lesions. The definition of a surgical cure varied among studies, with both an adequate decrease in intra-operative parathyroid hormone (IOPTH) and serum calcium/PTH normalization during follow-up used as criteria for defining a cure. Furthermore, the duration of postoperative follow-up to assess the normalization of serum calcium or PTH also varied.

Twenty-two studies used comparative imaging modalities such as ultrasound, MIBI, or 4D-CT [26,36,37,38,39,40,42,43,44,45,46,47,48,50,51,52,53,54,55,56,57,58]. Of these 22 studies, 3 studies used MRI as a second-line imaging modality after negative or discordant US and/or MIBI [36,39,47], and 2 studies investigated MRI as both a first-line and second-line imaging modality [45,57]. Two studies did not include comparative imaging modalities [41,49]. A total of 13 studies used a 1.5 T (PET-) MRI scanner [36,37,38,40,42,43,45,50,51,52,54,55,58], 10 studies used a 3.0 T (PET-) MRI scanner [26,39,41,44,46,47,49,53,56,57], and 1 study used both a 1.5 T and 3.0 T (PET-) MRI scanner [48]. All five studies that investigated PET/MRI used ^18^F-fluorocholine (^18^F-FCH) as the radioactive tracer [39,45,46,47,57].

Scanning protocols differed considerably between studies. In studies investigating 4D MRI, scanning protocols differed with regard to the amount of post-contrast phases and temporal resolution. Furthermore, MRI was performed by using a variety of techniques to obtain enhanced fat suppression, faster image acquisition, and reduced motion artifacts. For ^18^F-FCH-PET/MRI, the activity of ^18^F-FCH and the timing of image acquisition after the administration of ^18^F-FCH also varied. An overview of the reported scanning protocols and techniques is provided in Table 1.

### 3.1. Conventional MRI

Fourteen studies investigated conventional MRI for PHPT localization, with sensitivities ranging between 39.1% and 94.3% [37,38,39,40,42,43,45,48,51,53,54,55,56,58] (Table 2). Of the fourteen studies, two studies investigated conventional MRI for recurrent PHPT patients with prior parathyroid surgery with sensitivities of 82.0% and 76.9% [43,48]. Five studies reported specificity values of conventional MRI investigations, with specificities ranging from 50.0 to 100% [38,39,40,45,56].

### 3.2. 4D MRI

Eleven studies investigated 4D MRI and showed sensitivities ranging from 55.6% to 100% [26,36,40,41,44,47,48,49,50,52,53] (Table 2). Of these studies, nine investigated 4D MRI as a first-line imaging modality, with sensitivities varying from 64.0% to 100% [26,40,41,44,48,49,50,52,53]. The sensitivity of 4D MRI as second-line imaging after negative or discordant US and/or MIBI was investigated in two studies with reported sensitivities of 84.0% and 55.6% [36,47]. The sensitivity of 4D MRI in patients with recurrent PHPT with prior parathyroid surgery was investigated in two studies, with reported sensitivities of 93.3% and 90.1% [40,48]. A total of three MRI studies provided separate diagnostic values for patients with only MGD, with reported sensitivities ranging from 67.0% to 100% [26,41,49]. In a head-to-head comparative study, 4D MRI and 4D-CT were assessed as first-line imaging modalities, both demonstrating an identical sensitivity of 96.7% [52]. In a separate study, 4D MRI and 4D-CT were evaluated as second-line imaging modalities after negative or discordant US and MIBI. The sensitivity of 4D MRI was found to be 84.0%, while 4D-CT exhibited a sensitivity of 52.9%, although it is worth noting that not all patients underwent both imaging modalities [36]. Five studies reported specificity values of 4D MRI, with specificities varying from 66.0 to 100% [26,36,40,49,52]. Additionally, in five studies, the total 4D MRI scan time was less than 3 min [26,41,44,49,50].

### 3.3. ^18^F-FCH-PET/MRI

Five studies investigated ^18^F-FCH-PET/MRI, with sensitivities ranging between 84.2% and 100% [39,45,46,47,57] (Table 2). One study investigated ^18^F-FCH-PET/MRI as a first-line imaging modality, revealing a sensitivity of 84.2% [46]. Two studies investigated ^18^F-FCH-PET/MRI as a second-line imaging modality, with reported sensitivities of 100% and 90.0% [39,47]. Two studies utilized ^18^F-FCH-PET/MRI both as a first-line and second-line imaging modality, revealing a sensitivity of 100% and 88.9% [45,57]. Three studies reported specificity values of PET/MRI investigations, with specificities ranging from 96.0 to 100% [39,45,57]. The administered activity of ^18^F-FCH differed between studies and ranged from 93.8 to 350.0 MBq.

An overview of all diagnostic values reported in the included studies is available in Table 3. 

### 3.4. Quality Assessment

The results and interpretation of the quality assessment using the QUADAS-2 tool are shown in Appendix A. In most studies, an unclear or high risk of bias was found. Common sources of bias in these studies are selection bias, unclear reporting of blinding, absence of clinical follow-up to determine definitive curation, and use of different reference standards. Regarding the patient selection, a total of eight studies had a high risk of bias [37,40,45,48,49,51,54,55]. The main cause contributing to the high risk of bias regarding patient selection was the diversity among patients due to the presence of concomitant thyroid diseases (such as goiters) and a history of neck surgery. Furthermore, a total of 11 studies [36,38,41,42,43,44,46,50,52,53,58] had an unclear risk of bias with regard to patient selection due to a failure of reporting characteristics that may have influenced the diagnostic accuracy of the imaging modalities, such as the size of resected parathyroid adenomas, presence of concomitant diseases, and/or the presence of MGD in study groups. Regarding the index test, 12 studies [36,38,39,42,43,45,48,49,51,52,54,55] had an unclear risk of bias due to several causes, including inadequate reporting of blinding of radiologists. Regarding the reference test, three studies had a high risk of bias due to the utilization of different reference standards and the absence of clinical follow-up to determine the definitive curation [39,44,54]. The risk of bias was lowest for the flow and timing criterion, with only one study having a high risk of bias [39]. No major concerns were found regarding the applicability of the patient selection, index test, and reference test to the research question.

## 4. Discussion

This systematic review has provided an overview of studies that investigated the diagnostic accuracy of MRI in the localization of PHPT. The included studies used different MR imaging protocols, which can be categorized into three distinct groups: conventional MRI, 4D MRI, and PET/MRI. Conventional MRI showed sensitivities for PHPT localization ranging from 39.1% to 94.3%. More advanced MRI protocols, such as 4D MRI, demonstrated sensitivities between 55.6% and 100% for PHPT localization. Combining MR imaging with functional imaging techniques, like ^18^F-FCH-PET/MRI, yielded the highest overall diagnostic accuracy, with sensitivities ranging from 84.2% to 100%.

Fourteen of the studies included in this systematic review investigated the use of conventional MRI for localizing PHPT. The sensitivities observed in these studies ranged widely, spanning from 39.1% to 94.3%. Comparatively, a prior meta-analysis unveiled a pooled sensitivity of 83% for MIBI scintigraphy and 80% for ultrasound (US) imaging [59]. These results suggest that conventional MRI may not offer superior diagnostic accuracy when compared to MIBI and US. In a study by Hofer et al. [45], a sensitivity of 39.1% for conventional MRI was reported for the localization of PHPT. However, within the same study, when PET and MRI images were combined, the sensitivity increased significantly to 100%. As a result, the authors suggest that merging images from both modalities can enhance the precision of the anatomical localization [45]. Therefore, incorporating a PET/MRI protocol may be a more accurate imaging modality for the localization of PHPT.

Eleven of the included studies reported sensitivity rates of 4D MRI, which varied across a wide range, from 55.6% to a maximum of 100%. In nine of these studies, 4D MRI was used as a first-line imaging modality, achieving sensitivities within the range of 64% to 100% [26,40,41,44,48,49,50,52,53]. In comparison to 4D MRI, a recent meta-analysis compared 4D-CT and MIBI and found a sensitivity of 81.0% for 4D-CT and 65.0% for MIBI [16]. One of the included 4D MRI studies conducted a head-to-head comparison of 4D MRI and 4D-CT as first-line imaging modalities, revealing an identical sensitivity of 96.7% for both imaging modalities [52]. In another study included in this research, 4D MRI and 4D-CT were compared as second-line imaging modalities after negative or discordant US and MIBI. The sensitivity of 4D MRI was 84.0% compared to 52.9% for 4D-CT, although not all patients underwent both imaging modalities [36]. In previous research, it has been suggested that a higher number of post-contrast phases may result in a higher diagnostic accuracy [41,60]. One of the advantages of 4D MRI is that the number of post-contrast phases is not restricted by ionizing radiation, unlike 4D-CT, where every additional post-contrast phase adds to the total radiation exposure. Therefore, 4D MRI is able to analyze more post-contrast phases, which provides a more detailed analysis of the perfusion characteristics of parathyroid lesions [27]. One of the studies included in this review reported a rather low sensitivity of 55.6% when employing 4D MRI for the localization of PHPT. Notably, this study had a small sample size of only 10 patients who received 4D MRI scans with just two phases. These MRI scans were conducted following inconclusive ultrasound and/or MIBI results, potentially impacting the sensitivity compared to larger first-line imaging studies [47].

Five studies investigated the diagnostic accuracy of ^18^F-FCH-PET/MRI. These studies revealed an excellent diagnostic performance of 18F-FCH-PET/MRI for the localization of PHPT, with sensitivities ranging between 84.2% and 100% [39,45,46,47,57]. A recent network meta-analysis found that choline-PET-CT had the best diagnostic performance for localization of PHPT [61]. However, as the authors acknowledged, PET/MRI and PET-CT were not analyzed separately; PET/MRI studies were grouped with PET-CT studies. In four out of five studies in this systematic review, PET/MRI was used as a secondary imaging modality after initial US and/or MIBI scans yielded inconclusive or negative results. In these instances, localizing parathyroid glands is more challenging compared to patients who undergo imaging without prior inconclusive results. Furthermore, the use of ^18^F-FCH-PET/MRI would result in lower radiation exposure compared with ^18^F-FCH-PET-CT, making it a favorable alternative imaging modality, provided it offers similar diagnostic values.

In the re-operative setting where accurate preoperative localization is especially important, the sensitivity of traditional imaging modalities such as US and MIBI is reported to be as low as 52% and 54%, respectively [62]. In this review, two studies investigated 4D MRI and MIBI in patients with recurrent PHPT, demonstrating a superior diagnostic performance for 4D MRI, with reported sensitivities of 90.1% and 93.3%. [40,48]. In comparison, in a recent meta-analysis of 4D-CT, a pooled analysis of three studies revealed a sensitivity of 81% for 4D-CT in patients with recurrent PHPT [16].

Pre-operative imaging techniques generally have a lower diagnostic performance in patients with MGD compared to patients with SGD. This is primarily because solitary adenomas in SGD tend to be larger and, as a result, are more easily detectable than hyperplastic glands in MGD patients [63]. Three MRI studies provided separate diagnostic values for patients with MGD only, with sensitivities ranging from 67% to 100%. In patients with MGD, the literature reports the sensitivity of MIBI to be 44%, and a recent meta-analysis of 4D-CT found a sensitivity of 60% for the localization of MGD [16,64]. The observed sensitivities for MRI in MGD patients in this systematic review show promising results. However, more research is required to establish a more accurate estimate of the diagnostic accuracy of MRI in the detection of MGD compared to SGD. In studies with a large portion of MGD patients, the diagnostic accuracy of the imaging modality becomes lower, as MGD tends to be more difficult to correctly localize. Several studies in our paper included both SGD and MGD patients in their total cohort, although in varying proportions. Consequently, direct comparisons of sensitivities across studies become more challenging.

Overall, the decision on whether to use MRI for the preoperative localization of PHPT depends on the specific patient, the experience of the radiologists, costs, availability, and the individual institution’s imaging protocols [63,65]. The cost of (PET-) MRI can vary between health systems and scanners, and the difference with (PET-) CT may only be marginal in some institutions [30,66]. It is also important to note that the availability of (PET-) MRI scanners and 3T MRI scanners with multi-coil technology may be limited [48,67]. Cost-effectiveness studies regarding the costs of performing FCH PET/MRI for parathyroid localization are lacking. Given the superior sensitivity of FCH PET/MRI compared to both US and MIBI, it is imperative to conduct cost-effectiveness analyses to assess the viability of employing FCH PET/MRI for the detection of abnormal parathyroid glands.

The current literature has various limitations. Firstly, eleven of the studies employed a retrospective design with relatively small sample sizes. Additionally, most studies exhibited an unclear or high risk of bias. The majority of studies did not report specificity values. The specificity of a preoperative imaging modality is an essential parameter that may determine the success of a minimally invasive parathyroidectomy. Furthermore, there was inconsistency in sensitivity definitions among studies, with some utilizing patient-based sensitivity while others used lesion-based sensitivity. Most studies did not provide information on the curation rate, making it unclear whether surgical resection guided by MRI localization resulted in the cure of PHPT. This systematic review did not involve a meta-analysis due to considerable heterogeneity in the imaging protocols, making it unfeasible to pool the data in a meaningful quantitative analysis [68,69].

## 5. Conclusions

Recent technological advancements have enabled accurate and fast MR imaging. Despite the limitations of the available evidence, the results of this review indicate that the combination of MR imaging with functional techniques such as ^18^F-FCH-PET/MRI yielded the highest diagnostic accuracy. Further research on emerging MR imaging modalities, such as 4D MRI and PET/MRI, is warranted, as MRI exposes patients to minimal or no ionizing radiation compared to other imaging modalities.

## Figures and Tables

**Figure 1 diagnostics-14-00025-f001:**
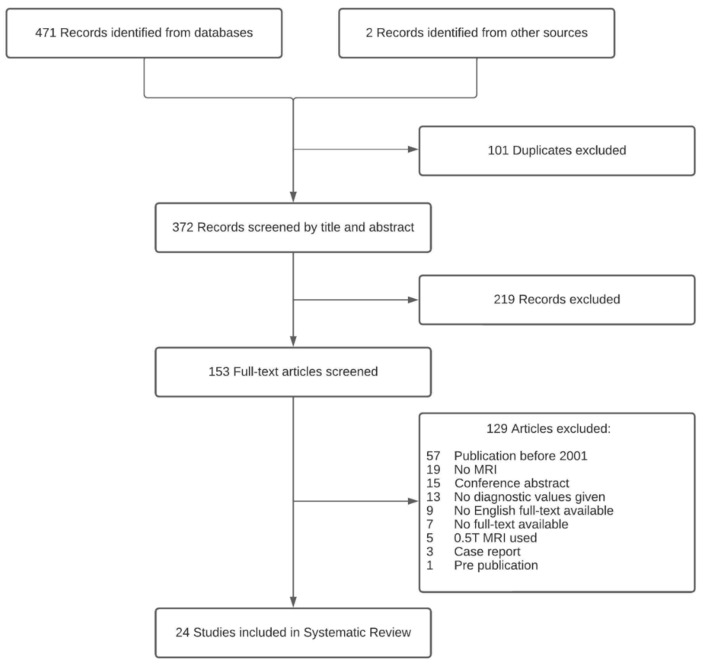
Flowchart: identification, screening, eligibility, and selection process.

**Table 1 diagnostics-14-00025-t001:** Characteristics of all studies included in the qualitative analysis.

Study	Study Design	Indication	No. of Patients	SGD/MGD	MRI Technique	4D MRI Post-Contrast Phases	4D MRI Temporal Resolution (Seconds)	MRI Sequence Techniques	18F-FCH PET Protocol and 18F-FCH Activity (MBq)	MRI Scanner
Agha et al., 2012 [37]	Prospective	PHPT	30	29/1	Conventional MRI (CE)	NR	NR	TSE STIR BLADE	NR	1.5 T Magnetom Symphony (Siemens©)
Akbaba et al., 2011 [38]	Prospective	PHPT	94	92/2	Conventional MRI (NC)	NR	NR	FSE STIR Respiratory compensation	NR	1.5 T Signa (General Electric©)
Bijnens et al., 2022 [56]	Retrospective	PHPT	104	97/5	Conventional MRI (CE)	NR	NR	NR	NR	3 T Ingenia or Ingenia Elition (Philips©)
Cakal et al., 2012 [42]	Prospective	PHPT	39	38/1	Conventional MRI (NC)	NR	NR	STIR	NR	1.5 T Achieva (Philips©)
Gotway et al., 2001 [44]	Retrospective	Recurrent/persistent PHPT	119	NR	Conventional MRI (CE)	NR	NR	FSE	NR	1.5 T Signa (General Electric©)
Michel et al., 2013 [51]	Prospective	PHPT	58	NR	Conventional MRI (NC)	NR		STIR	NR	1.5T Magnetom Symphony (Siemens©)
Ruf et al., 2004 [58]	Prospective	PHPT	17	16/1	Conventional MRI (CE)	NR	NR	NR	NR	1.5 T Magnetom SP63 (Siemens©)
Saeed et al., 2006 [54]	Retrospective	PHPT	26	NR	Conventional MRI (CE)	NR		FSE	NR	1.5 T Signa (General Electric©)
Sekiyama et al., 2003 [55]	Retrospective	PHPT	75	64/11	Conventional MRI (CE)	NR	NR	STIR	NR	1.5 T Sigma Horizon (General Electric©)
Aschenbach et al., 2012 [40]	Retrospective	Recurrent/persistent PHPT	30	30/0	4D MRI and Conventional MRI (NC)	10	6	FSE STIR	NR	1.5 T Signa Excite II (General Electric©)
Kluijfhout et al., 2016 [48]	Retrospective	PHPT + recurrent/ persistent PHPT	125	114/9	4D MRI and Conventional MRI (CE)	4	NR	FSE	NR	1.5 T and 3.0 T Scanner not specified
Ozturk et al., 2019 [53]	Retrospective	PHPT	41	40/1	4D MRI and Conventional MRI (NC)	8	15	TSE STIR Dixon VIBE	NR	3 T Ingenia (Philips©)
Acar et al., 2020 [37]	Retrospective	PHPT undetected ^a^	25	24/1	4D MRI	NR	13	NR	NR	1.5 T Avanto (Siemens©)
Argiro et al., 2018 [26]	Prospective	PHPT	56	49/8	4D MRI	6	18	IDEAL FSE	NR	3 T Discovery MR 750 (General Electric©)
Becker et al., 2020 [41]	Prospective	PHPT	54	37/17	4D MRI	24	6	Dual-echo for Dixon fat suppression TWIST CAIPIRINHA	NR	3T Magnetom Skyra MR (Siemens©)
Grayev et al., 2012 [44]	Prospective	PHPT	25	24/1	4D MRI	20	5.4	FSE SPGR TRICKS IDEAL	NR	3T Twin Speed V12.0 and V14.0 (General Electric©)
Memeh et al., 2019 [49]	Retrospective	PHPT	26	18/8	4D MRI	NR	NR	TWIST CAIPIRINHA DIXON	NR	3T Skyra (Siemens©)
Merchavy et al., 2016 [50]	Prospective	PHPT	11	NR	4D MRI	10	13	TSE MEDIC TWIST STIR VIBE	NR	1.5-T Espree (Siemens©)
Murugan et al., 2021 [52]	Prospective	PHPT	48	41/7	4D MRI	NR	NR	TSE BTFE DIXON	NR	1.5 T Achieva (Philips©)
Hofer et al., 2020 [45]	Retrospective	PHPT + PHPT undetected ^a^	42	38/4	18F-FCH-PET-fusion- (4D-) MRI (CE) and Conventional MRI	NR	NR	STIR	Single acquisition: 60 min Activity: 250–350	1.5 T Avanto (Siemens©)
Khafif et al., 2019 [46]	Prospective	PHPT	19	19/0	18F-FCH PET/(4D)-MRI	20	6	DIXON VIBE	Dynamic: 16 min Activity: 93.75	3.0 T PET/MRI, Biograph mMR (Siemens©)
Kluijfhout et al., 2017 [47]	Prospective	PHPT undetected ^a^	10	8/2	18F-FCH-PET/(4D)-MRI	NR	12	FSE, IDEAL, STIR	Dynamic: 40 min Activity: 188	3.0 T PET/MR (General Electric©)
Araz et al., 2021 [39]	Retrospective	PHPT undetected ^a^	36	NR	18F-FCH-PET/MRI (NC)	NR	NR	TSE STIR	Single acquisition: 45–60 min Dynamic: 20 min Activity: 100	3.0 T Signa PET/MR (General Electric©)
Graves et al., 2022 [57]	Prospective	PHPT + Recurrent/persistent PHPT + undetected ^a^	101	NR	18F-FCH-PET/MRI (NC)	NR	NR	NR	Single Acquisition: 20–60 min Activity: 148–259	3.0 T PET/MRI (General Electric©)

Abbreviations: 4D MRI: dynamic-enhanced magnetic resonance imaging, 18F-FCH; 18F-fluorocholine, CAIPIRINHA: Controlled Aliasing in Parallel Imaging Results in Higher Acceleration, FSE; fast-spin echo, IDEAL; iterative decomposition of water and fat with echo asymmetry and least-squares estimation, MBq: megabecquerel, MGD: multigland disease, MEDIC: Multiple Echo Data Image Combination, NR: not reported, PET; positron emission tomography, PHPT: primary hyperparathyroidism, SGD: single-gland disease, SPGR: spoiled gradient recalled echo, STIR; Short-TI Inversion Recovery, T; Tesla, TSE; turbo spin echo, TRICKS; Time-Resolved Imaging of Contrast Kinetics, TWIST: Time-Resolved Angiography With Stochastic Trajectories, CE: contrast-enhanced, NC: non-contrast. ^a^ Undetected: after negative US and/or 99mTc-sestamibi.

**Table 2 diagnostics-14-00025-t002:** Reported sensitivities of all studied imaging modalities in the included studies.

Lesion-Based	Sensitivity							
Study	Indication	Conventional MRI (%)	4D MRI (%)	PET/MRI (%)	US (%)	Sestamibi (%)	4D-CT (%)	Choline PET/CT (%)
Agha et al., 2012 [37]	PHPT	71.0 (22/31)	NR	NR	74.2 (23/31) 100 ^b^ (31/31)	81.0 (25/31) NR	NR	NR
Akbaba et al., 2011 [38]	PHPT	63.8 (60/94)	NR	NR	87.2 (82/94)	75.5 (71/94)	NR	NR
Bijnens et al., 2022 [56]	PHPT	60.0 (12/20)	NR	NR	75.0 (104/139)	57.0 (55/97)	90.0 (30/33)	NR
Cakal et al., 2012 [42]	PHPT	66.7 (26/39)	NR	NR	89.7 (35/39)	71.8 (28/39)	NR	NR
Gotway et al., 2001 [43]	Recurrent/persistent PHPT	82.0 (107/30)	NR	NR	NR	85.0 (110/130)	NR	NR
Ruf et al., 2004 [58]	PHPT	71.0 (10/14)	NR	NR	NR	86.0 (12/14)	NR	NR
Sekiyama et al., 2003 [55]	PHPT	73.0 (43/59)	NR	NR	70.0 (40/63)	78.0 (21/27)	NR	NR
Kluijfhout et al., 2016 [48]	PHPT + recurrent/ persistent PHPT	Total: 81.1 (77/95) PHPT: 76.9 (20/26) Recurrent PHPT: 82.4 (56/68)	Total: 78.8 (35/45) PHPT: 65.2 (15/23) Recurrent PHPT: 90.1 (20/22)	NR	Total: 51.1 (67/131) PHPT: 45.7 (21/46) Recurrent PHPT: 54.1 (46/85)	Total: 68.8 ^c^ (86/125) PHPT: 60.0 ^c^ (21/35) Recurrent PHPT: 74.2 ^c^ (66/89)	NR	NR
Ozturk et al., 2019 [53]	PHPT	NR	90.5 (38/42)	NR	76.2 (32/42)	71.4 (30/42)	NR	NR
Acar et al., 2020 [36]	PHPT undetected ^a^	NR	84.0 (21/25)	NR	NR	NR	52.9 (9/17)	NR
Argiro et al., 2018 [26]	PHPT	NR	97.8 (45/46) MGD: 100 (8/8)	NR	89.1 (41/46)	83.6 (38/46)	NR	NR
Becker et al., 2020 [41]	PHPT	NR	83.3 (70/84) SGD: 92.0 (34/37) MGD: 77 (36/47)	NR	NR	NR	NR	NR
Grayev et al., 2012 [44]	PHPT	NR	64.0 (16/25)	NR	88.9 (8/9)	72.0 (18/25)	NR	NR
Memeh et al., 2019 [49]	PHPT	NR	SGD: 100 (14/14) MGD: 67.0 (8/12)	NR	NR	NR	NR	NR
Murugan et al., 2021 [52]	PHPT	NR	96.7 (59/61)	NR	NR	NR	96.7 (59/61)	NR
Hofer et al., 2020 [45]	PHPT + PHPT undectected ^a^	39.1 (18/46)	NR	100 (46/46)	NR	NR	NR	93.5 (43/46)
Michel et al., 2013 [51]	PHPT	94.3 (53/56)	NR	NR	NR	88.0 (47/54)	NR	NR
Saeed et al., 2006 [54]	PHPT	73.0 (19/26)	NR	NR	NR	65.4 (17/26)	NR	NR
Aschenbach et al., 2012 [40]	Recurrent/persistent PHPT	63.3 (19/30)	93.3 (28/30)	NR	NR	80.0 (24/30)	NR	NR
Merchavy et al., 2016 [50]	PHPT	NR	91.0 (10/11)	NR	91.0 (10/11)	91.0 (10/11)	NR	NR
Kluijfhout et al., 2017 [47]	PHPT undetected ^a^	NR	55.6 (5/9)	90.0 (9/10)	NR	NR	NR	NR
Araz et al., 2021 [39]	PHPT undetected ^a^	80.0 (24/30)	NR	100 (30/30)	NR	NR	NR	NR
Graves et al., 2022 [57]	PHPT + recurrent/persistent PHPT + undetected ^a^	NR	NR	88.9 (64/72)	37.1 (26/70)	27.5 (19/69)	NR	NR
Khafif et al., 2019 [46]	PHPT	NR	NR	84.0 (16/19)	84.0 (16/19)	74.0 (14/19)	NR	NR

Abbreviations: 4D MRI: dynamic-enhanced magnetic resonance imaging, PHPT; primary hypeparathyroidism, 4D-CT; dynamic contrast-enhanced computed tomography, NR: not reported, MRI: magnetic resonance imaging, PET/CT; positron emission computed tomography, US: ultrasound, SGD: single-gland disease, MGD: multigland disease. ^a^ Undetected: After negative US and/or 99mTc-sestamibi, ^b^ CEUS: Contrast-Enhanced Ultrasound, ^c^ Patients were studied either using planar imaging or single-photon emission CT (SPECT) imaging.

**Table 3 diagnostics-14-00025-t003:** All reported diagnostic values of included MRI studies.

**Lesion-Based **						
**Study**	**Type of MRI Protocol**	**Sens %**	**Spec %**	**NPV %**	**PPV %**	**Accuracy %**
Agha et al., 2012 [37]	Conventional MRI	71.0 (22/31)	NR	NR	NR	NR
Akbaba et al., 2011 [38]	Conventional MRI	63.8 (60/94)	50.0 (2/4)	5.5 (2/36)	96.8 (60/62)	63.3 (62/98)
Bijnens et al., 2022 [56]	Conventional MRI	60.0 (12/20)	83.3 (5/6)	NR	NR	NR
Cakal et al., 2012 [42]	Conventional MRI	66.7 (26/39)	NR	NR	100 (26/26)	NR
Gotway et al., 2001 [43]	Conventional MRI	82.3 (107/130)	NR	NR	89.2 (116/130)	NR
Ruf et al., 2004 [58]	Conventional MRI	71.0 (10/14)	NR	NR	NR	NR
Sekiyama et al., 2003 [55]	Conventional MRI	72.9 (43/59)	NR	NR	NR	NR
Kluijfhout et al., 2016 [48]	Conventional MRI 4D MRI	81.1 (77/95) 78.8 (35/45)	NR NR	NR NR	84.6 (77/91) 87.5 (35/40)	NR NR
Ozturk et al., 2019 [53]	Conventional MRI 4D MRI	81.0 (34/42) 90.5 (38/42)	NR NR	NR NR	87.2/91.9 95.0 (38/40)	NR NR
Acar et al., 2020 [36]	4D MRI	84.0 (21/25)	97.4 (73/75)	94.8 (73/77)	91.3 (21/23)	94.0 (21/25)
Argiro et al., 2018 [26]	4D MRI	97.8 (45/46)	97.5 (118/121)	99.2 (/118/119)	93.7 (45/48)	NR
Becker et al., 2020 [41]	4D MRI	Overall—83.3 (70/84) SGD—92.0 (34/37) MGD—77.0 (36/47)	NR	NR	NR	NR
Grayev et al., 2012 [44]	4D MRI	64.0 (16/25)	NR	NR	67.0 (16/24)	NR
Memeh et al., 2019 [49]	4D MRI	100 (14/14)	66.0 (8/12)	NR	NR	84.6 (22/26)
Murugan et al., 2021 [52]	4D MRI	96.7 (59/61)	66.6 (2/3)	50.0 (2/4)	98.3 (59/60)	95.2 (58/61)
Hofer et al., 2020 [45]	Conventional MRI PET/MRI	39.1 (18/46) 100 (46/46)	89.3 (109/122) 100 (122/122)	79.6 (109/137) 100 (122/122)	58.1 (19/32) 100 (46/46)	75.6 (92/122) 100 (122/122)
**Patient-Based**						
**Study**	**Type of MRI protocol**	**Sens %**	**Spec %**	**NPV %**	**PPV %**	**Accuracy %**
Michel et al., 2013 [51]	Conventional MRI	94.3 (53/56)	NR	NR	96.2 (53/55)	NR
Saeed et al., 2006 [54]	Conventional MRI	73.1 (19/26)	NR	NR	95.0 (19/20)	NR
Aschenbach et al., 2012 [40]	Conventional MRI 4D MRI	63.3 (19/30) 93.3 (28/30)	100 (30/30) 100 (30/30)	NR	NR	NR
Merchavy et al., 2016 [50]	4D MRI	91.0 (10/11)	NR	NR	NR	NR
Kluijfhout et al., 2017 [47]	PET/MRI 4D MRI	90.0 (9/10) 55.6 (5/9)	NR NR	NR NR	100 (10/10) NR	NR NR
Araz et al., 2021 [39]	PET/MRI Conventional MRI	100 (30/30) 80.0 (24/30)	100 (30/30) 50.0 (4/8)	100 (30/30) 64.0 (7/11)	100 (30/30) 70.0 (21/30)	100 (30/30) 68.0 (23/34)
Graves et al., 2022 [57]	PET/MRI	88.9 (64/72)	96.0 (97/101)	NR	NR	NR
Khafif et al., 2019 [46]	PET/MRI	84.2 (16/19)	NR	NR	NR	NR

Abbreviations: 4D MRI: dynamic-enhanced magnetic resonance imaging, NR: not reported, MRI: magnetic resonance imaging, SGD: single-gland disease, and MGD: multigland disease.

## Data Availability

The datasets produced or examined in the course of this study have been incorporated into the published article.

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
