# Peer review of "Diagnostic Performance of Magnetic Resonance Imaging for Parathyroid Localization of Primary Hyperparathyroidism: A Systematic Review"

_diagnostics, 2023, doi:10.3390/diagnostics14010025_

Round 1

Reviewer 1 Report

Comments and Suggestions for Authors

Dear Authors,

Thank you for this very interesting paper summarizing MRI techniques used in parathyroid gland imaging.

Please correct "dose" to "activity" when referring to the administered activity of radiopharmaceuticals.

Please add data on the cost-effectiveness of all mentioned MRI techniques, and compare them to other techniques used in parathyroid gland imaging.

Author Response

Dear Reviewer,

We would like to express our sincere appreciation for your thoughtful review of our manuscript entitled "Diagnostic Performance of Magnetic Resonance Imaging for Parathyroid Localization of Primary Hyperparathyroidism". We are grateful for your positive feedback and constructive suggestions, which have undoubtedly contributed to the enhancement of the overall quality of our work.

In response to your insightful comments, we have carefully reviewed the manuscript and are committed to addressing the specific points raised. Specifically, we acknowledge the need to rectify the terminology used and will duly replace "dose" with "activity" when referring to the administered activity of radiopharmaceuticals.

Furthermore, we appreciate your suggestion to incorporate data on the cost-effectiveness of the MRI techniques discussed in our paper. In light of this, we will include relevant information on the available cost-effectiveness data of MRI techniques for the localization of parathyroid adenomas.

We believe that these revisions will significantly strengthen the comprehensiveness and accuracy of our manuscript. Once these modifications are completed, we will promptly submit the revised manuscript for your further consideration.

Thank you once again for your valuable input and for guiding us toward an improved and more robust presentation of our research. We are dedicated to ensuring that our manuscript meets the high standards of “Diagnostics”, and we look forward to the opportunity to share our refined work with you.

Kind regards

Reviewer 2 Report

Comments and Suggestions for Authors

Dear Authors

I read and checked your article carefully. The title is elegant, the abstract is sufficient, the introduction is well written. A systemic review was written about the place and guidance of magnetic resonance imaging in the diagnosis of primary hyperparathyroidism. Pubmed and Embase servers were searched and the articles matching the keywords were reviewed by the authors and the articles that met the inclusion criteria were examined. Exclusion criteria were also clearly stated in the article. When I searched Pubmed, I did not encounter such a comprehensive review written for this purpose. 24 articles that met the inclusion criteria were reviewed. The findings were very detailed and detailed information was provided under subheadings according to the imaging method. The tables and figures are very high quality, legible and understandable. The sensitivity of different MR techniques in detecting parathyroid lesions in patients with primary hyperparathyroidism was investigated and these techniques were compared with previous studies in the discussion. Limitations of the study are also discussed in detail. The importance of magnetic resonance imaging in this group of patients is mentioned because it does not involve ionizing radiation. The English used is easy to understand, I did not find any spelling/typing errors. The references are up to date and conform to the journal format.

Author Response

Dear Reviewer,

Thank you for your thorough and insightful review of our manuscript titled "Diagnostic Performance of Magnetic Resonance Imaging for Parathyroid Localization of Primary Hyperparathyroidism".  We sincerely appreciate your positive feedback and constructive comments, which have greatly contributed to the refinement of our work.

We will carefully consider all your suggestions to further enhance the manuscript's quality. Your valuable insights have been instrumental in shaping a more robust and impactful contribution to the field. We are dedicated to addressing any additional recommendations you may have and look forward to resubmitting the revised manuscript for your further evaluation.

Thank you once again for your time, attention, and invaluable feedback.

Best regards